# Adapting a Clinical Practice Guideline for Management of Patients with Knee and Hip Osteoarthritis by Hong Kong Physiotherapists

**DOI:** 10.3390/healthcare11222964

**Published:** 2023-11-15

**Authors:** Fadi M. Al Zoubi, Arnold Y. L. Wong, Gladys L. Y. Cheing, Jason P. Y. Cheung, Siu Ngor Fu, Helen H. L. Tsang, Rainbow K. Y. Law, Billy Chun Lung So, Raymond Tsang, Sharon Tsang, Chunyi Wen, Michael Wong, Yim Ching Yau, André E. Bussières

**Affiliations:** 1Department of Rehabilitation Sciences, The Hong Kong Polytechnic University, Hong Kong SAR, China; arnold.wong@polyu.edu.hk (A.Y.L.W.); gladys.cheing@polyu.edu.hk (G.L.Y.C.);; 2Department of Orthopaedics and Traumatology, The University of Hong Kong, Pokfulam, Hong Kong SAR, China; cheungjp@hku.hk; 3Department of Medicine, The University of Hong Kong, Hong Kong SAR, China; 4Physiotherapy Centre, Hong Kong Sanatorium & Hospital, Hong Kong SAR, China; 5Hong Kong Physiotherapy Association, Hong Kong SAR, China; 6Physiotherapy Department, MacLehose Medical Rehabilitation Centre, Hong Kong SAR, China; 7Department of Biomedical Engineering, The Hong Kong Polytechnic University, Hong Kong SAR, China; 8Rehabilitation Clinic, The Hong Kong Polytechnic University, Hung Hom, Kowloon, Hong Kong SAR, China; 9Nursing Mixed Surgical Ward, Hong Kong Sanatorium & Hospital, Hong Kong SAR, China; connie.yau@connect.polyu.hk; 10School of Physical and Occupational Therapy, Faculty of Medicine and Health Sciences, McGill University, Montreal, QC H3G 2M1, Canada; andre.bussieres@mcgill.ca; 11Département Chiropratique, Université du Québec à Trois-Rivières, Trois-Rivières, QC G8Z 4M3, Canada

**Keywords:** practice guideline, practice guidelines as topic, adaptation study, ADAPTE framework, osteoarthritis, hip, osteoarthritis, knee, physical therapy modalities

## Abstract

Knee and hip osteoarthritis are common disabling conditions globally. Although numerous international clinical practice guidelines exist to guide physiotherapy management, not all recommendations issued from these guidelines can be translated to other contexts without considering the cultural acceptability and clinical implementability of targeted countries. Because the ADAPTE framework provides a robust methodology to adapt guidelines to the local context, this study used its methodology to adapt high-quality guideline recommendations to promote optimal physiotherapy care for knee and hip osteoarthritis in Hong Kong. The ADAPTE framework was used and modified to complete the adaptation process. International clinical practice guidelines were identified from eight guideline clearinghouses and six electronic databases. Two independent reviewers critically appraised the eligible guidelines using the AGREE II tool. We extracted and tabulated recommendations from high-quality guidelines. A voting-based consensus among interdisciplinary experts was conducted to decide on suitable recommendations for the Hong Kong context and whether there was a need to modify them. Pertinent recommendations were then translated into the traditional Chinese language. Our team members suggested modifying four tools and adding one to explore the patient’s feedback on the recommendations, to the ADAPTE framework. The adaptation was performed on three high-quality guidelines. We adapted 28 and 20 recommendations for treating knee and hip osteoarthritis, respectively. We recommend a multimodal treatment for managing knee and hip osteoarthritis. Land- and aquatic-based exercises, patient education, and self-management were strongly recommended for patients with knee osteoarthritis. Land- and aquatic-based exercises were strongly recommended for patients with hip osteoarthritis. This is the first adaptation study in Hong Kong. It provides guidance to local physiotherapists on managing patients with knee and hip osteoarthritis. Future studies should test the effectiveness of implementing this adapted guideline to improve local physiotherapy care in Hong Kong.

## 1. Introduction

Osteoarthritis (OA) is one of the most prevalent and disabling health conditions globally, affecting over 595 million patients in 2020 [1]. The prevalence of joints with OA differs across geographical regions. For instance, the age-standardized prevalence of knee and hip OA is the highest in North America and the high-income Asia Pacific region, while it is the lowest in Sub-Saharan Africa [2]. A 2020 umbrella review found that knee and hip OA were associated with overweight/obese, older age, females, low levels of strength or fitness, athletic competition-related joint injuries, heavy lifting at the workplace, climbing, squatting, and kneeling [3]. 

In Hong Kong (HK), the prevalence of OA was greatest in the knee, followed by the hip [4]. In addition to previously reported risk factors for developing knee and hip OA [3], people in HK with lower education and socioeconomic status have more severe OA conditions [5,6,7]. The estimated direct and indirect costs of OA have caused significant individual, societal, and economic burdens in HK [8]. The direct costs of OA are similar to those in high-income countries; however, this burden is predominantly covered by the government [5].

Globally, it has been estimated that over 344 million patients with OA need rehabilitation services, including non-pharmacological management provided by physiotherapists [9]. To reduce ineffective or unsafe practices, physiotherapists are expected to adopt evidence-based practice approaches in their management of musculoskeletal conditions [10]. Evidence-based clinical practice guidelines (CPGs) aim to improve the effectiveness of healthcare services while reducing costly and undesirable practices [11]. 

To date, only two Chinese CPGs have been published on the management of knee and hip OA. The first, now outdated, was developed in 2004 [12], targeting family doctors without physiotherapy-related recommendations. The second more generic CPG was published in 2019 [13], targeting all clinicians across all Chinese regions. While it contained recommendations for all osteoarthritic joints, this more recent CPG recommended physiotherapy as a non-pharmacological therapeutic option instead of a distinct profession that can deliver a broad spectrum of therapeutic interventions. Thus, these two CPGs are not appropriate to guide the physiotherapy practice in HK.

There are three commonly used approaches for developing CPGs: creating a new guideline, adopting an existing high-quality CPG as is, or adapting high-quality CPGs while taking into account the cultural, clinical, and organizational contexts of the local community [14,15].

Adapting CPG may better suit the HK context, given the cultural and clinical variations between Western and HK rehabilitation practices. For instance, HK physiotherapy schools teach acupuncture. Therefore, a CPG recommending the use of acupuncture to manage patients with OA would likely be perceived as a facilitator among HK physiotherapists but as a possible barrier for the majority of Western physiotherapists. Guideline adaptation represents a paramount solution to avoid work duplication and time-wasting [16].

There are many frameworks that have been used to adapt CPGs to the local context. Our team selected the ADAPTE framework [15] as it is the most developed and used process which has been used in different cultures and countries [17,18,19,20,21,22,23,24,25,26,27,28,29,30,31,32]. In addition, this framework has a toolkit with a user manual to facilitate the use of each step [33].

Against this background, the current study had two objectives. First, to describe the adaptation methods using the ADAPTE framework and its toolkit to adapt CPG to the HK context. Second, to improve the management of knee and hip OA among HK physiotherapists by adapting relevant CPG recommendations on knee and hip OA treatments.

## 2. Materials and Methods

This study used the ADAPTE framework (version 2.0) [15] and the Adaptation Resource Toolkit provided by the Guideline International Network [33]. The ADAPTE framework consists of 24 steps distributed in 3 phases: setup, adaptation, and finalization. 

### 2.1. Phase 1: Setup

An organizing committee consisting of three people (AB, AW, and FAZ) was formed to oversee the entire ADAPTE process. This committee consisted of methodologists with expertise in developing and implementing CPGs, conducting systematic reviews and quality appraisals of CPGs, and two members familiar with physiotherapy practice in HK. The committee focused on recommendations related to the non-pharmacological therapeutic options routinely delivered by physiotherapists for the management of ‘knee and hip OA’ because of its high prevalence among HK people [4]. Therefore, pharmacological and surgical interventions were excluded, as these interventions are not within the scope of the practice of physiotherapists.

To check whether the adaptation was feasible, we first conducted an exploratory search of the literature. While several recent CPGs and overviews of CPGs for the management of OA [34,35,36] were identified, the search failed to uncover CPGs focusing on non-pharmacological interventions that can be provided by physiotherapists in managing knee and hip OA in China or HK.

A guideline adaptation panel was tasked with the adaptation process. The panel was composed of 13 stakeholders, with expert clinicians having personal experience in managing patients with OA; policy, administrative, or management expertise; researchers with methodological expertise; critical appraisal of the literature; implementation science; and HK culture. Specifically, this multidisciplinary panel included musculoskeletal physiotherapists (RL, RT, MW), physiotherapy researchers (FAZ, AW, HC, AF, BS, ST), rheumatologists (HT, JW), orthopedists (JC, CW), a nurse (CY), as well as a representative from the HK Physiotherapy Association (RT). The views and preferences of patients with knee and hip OA were explored after formulating the recommendations. 

### 2.2. Phase 2: Adaptation

#### 2.2.1. Scope and Purpose Module

In this phase, we used the PIPOH format (Population, Interventions, Professionals, Outcomes, and Healthcare settings) [33] to identify a specific health question for this guideline. Our research question was: What are the conservative non-pharmacological treatments that can be recommended for the management of patients with knee and hip OA and provided by HK physiotherapists? 

#### 2.2.2. Search and Screen Module 

After formatting the research question, literature search strategies were developed with the consultation of a health librarian (see Appendix A). We searched six key electronic databases: MEDLINE (PubMed), EMBASE, CINHAL (Complete EbscoHost), PEDro, Scopus, and Epistemonikos. Additionally, we searched several guidelines clearinghouses, but only a few of them were still updating their guidelines lists. Table 1 presents the full list of electronic databases and guideline clearinghouses that were searched. These databases were first searched on 1 June 2021 and again on 1 November 2021. 

We included CPGs that were (a) published within the past 5 years (2016–2021), (b) available in English or Chinese, (c) designed based on systematic reviews that answered specific research questions, and (d) focused on conservative non-pharmacological management options that are within the scope of physiotherapy practice for managing knee and hip OA in all healthcare settings. We excluded CPGs that were (a) targeting the pediatric population, (b) based on consensus, and (c) focusing on surgical or pharmaceutical interventions.

Two reviewers (FAZ and a trained research assistant) independently applied the eligibility criteria to screen titles and abstracts. Reviewers resolved any disagreements through discussions, and a third reviewer (AB) adjudicated any persistent disagreements. 

#### 2.2.3. Quality Assessment Module 

All included CPGs were critically appraised for methodological quality by two independent appraisers (FAZ and AW) using the Appraisal of Guidelines for Research and Evaluation (AGREE) II tool [37]. The AGREE II tool is a reliable and valid evaluation tool containing 23 items in 6 domains (scope and practice, stakeholder involvement, rigor of development, clarity of presentation, applicability, and editorial independence) [37]. Each item of the AGREE II tool was rated on a 7-point Likert scale, ranging from 1 (strongly disagree) to 7 (strongly agree). The domain scores range from 0 to 100%, with higher scores representing stronger between-reviewer agreements. The domain scores were calculated using the formulas in the AGREE II manual [38]. To identify a guideline of high quality, the organizing committee proposed two criteria: (a) a cut-off score ≥ 60% for at least 4 out of 6 of the AGREE II domains; and (b) a score of ≥75% for the domain ‘methodological rigor’. The two appraisers initially appraised the included CPGs. If the discrepancy in the rating of any individual item was ≥2, a discussion was held until consensus was reached.

#### 2.2.4. Extracting Recommendations from High-Quality CPGs 

We extracted recommendations only from eligible, high-quality CPGs. Recommendations judged to be unrelated to knee and hip OA were excluded. Two independent reviewers (FAZ and ST) categorized high-quality CPGs’ recommendations into three groups: non-pharmacological treatments mainly provided by HK physiotherapists (✓), non-pharmacological treatments partly provided by HK physiotherapists as the intervention requires certain postgraduate training (?), and other treatments not provided by HK physiotherapists/not within the scope of practice (✗). Only recommendations that were judged by both reviewers to be (✓) or (?) were included. 

Next, the similarities and differences across recommendations from the identified high-quality guidelines were tabulated in matrices. These matrices were presented to the guideline adaptation panel to assess the applicability (e.g., organizational and system barriers, treatment availability, and resource availability in HK) and acceptability (e.g., whether the recommendations are compatible with HK culture and values) of the recommendations.

Members of the guideline adaptation panel attended an online meeting in November 2021. Prior to the meeting, the panel committee received the working plan, the adaptation process, all eligible high-quality guidelines, and the matrices. During the meeting, two members of the organizing committee (FAZ and AW) encouraged the discussion among the panel members to reach a consensus about the feasibility, implementability, and applicability of the recommendations in HK. Panel members then received a link to anonymously vote on each guideline using a 3-point scale (accept as is, accept with modification, reject this guideline). Members were informed a priori that at least 75% of the panel had to vote and reach an 80% agreement for a given recommendation to be retained. If no consensus was reached, we encouraged further discussion among panelists prior to conducting additional rounds (*n* = 1–2) of voting to reach an agreement. If the panel decided that one of the guidelines was superior to others, our plan was to contact the original authors and ask for permission to adapt their guideline recommendations. If two or more guidelines were deemed acceptable, we planned to invite the panel to consider each recommendation in terms of quality, strength, wording, and relevance to the HK practice and then vote anonymously on the recommendations using an online tool (https://strawpoll.com/, accessed on 17 November 2022). 

All these steps were recorded, and all modifications to the recommendations were documented. The final draft of the adapted guideline provided an account of the steps taken to reach an agreement on recommendations, along with related documentation. The statements generated by the committee were categorized as strongly recommended, moderately recommended, conditionally recommended, neutral, conditionally against, moderately against, and strongly against.

### 2.3. Phase 3: Finalization

The final draft of the adapted guideline was sent to an external review panel composed of all relevant stakeholders, including local researchers, practicing physiotherapists, policymakers, and experts from professional bodies. A purposive sampling method was employed to recruit a total of 10–15 external reviewers from different healthcare settings and representatives of multiple HK geographical areas. This final phase allowed for the formal endorsement of the guideline by key local stakeholders. Endorsement by professional bodies may boost guideline uptake among their members. External reviewers were encouraged to provide their comments, feedback, and appraisals of the adapted CPG using a survey. 

In addition, we translated the recommendations to Chinese by two members of this guideline and sent them to a convenient sample of five patients with knee OA and three with hip OA to obtain their feedback. Patients were recruited from our team members’ networks and encouraged to read the recommendations and comment on the perceived applicability and feasibility of implementing those recommendations. The feedback received from the external reviewers and patient samples was considered by the organizing committee, which addressed each comment and further refined the adapted CPG. The final version of the adapted CPG was submitted for publication in a peer-reviewed journal and presented at national and international conferences.

## 3. Results

### 3.1. Phase 1: Setup

The ADAPTE framework started by checking whether the adaptation was feasible, followed by establishing a committee, and then selecting a topic. The order of these steps is not rational, as we need to first establish a committee, then select the topic, and finally check the feasibility of adaptation.

### 3.2. Phase 2: Adaptation

#### Search and Screen Module 

Figure 1 presents the PRISMA flowchart illustrating the results of searching, screening, and full-text reviewing. Our search retrieved a total of 8068 citations from the six electronic databases. After removing the duplicates (*n* = 3894), 4174 titles and abstracts were screened, resulting in the exclusion of 4134 citations. Thirty-one CPGs were excluded with reasons after the full-text reviews, yielding nine eligible guidelines [39,40,41,42,43,44,45,46,47] (Appendix A). Our search on the guideline clearinghouse websites identified 36 guidelines, of which 31 were ineligible and 5 were included [48,49,50,51,52] (Appendix A). 

In total, we included 14 publications representing 12 guidelines because the Ottawa guideline for knee OA was published in 3 separate publications [43,44,45]. Three guidelines only targeted knee OA and were developed by the American Academy of Orthopedic Surgeons (AAOS) [52], the Turkish League Against Rheumatism (TLAR) [39], and the Ottawa panel [43,44,45]. Three guidelines targeted hip OA alone and were developed by the AAOS [51], the American Physical Therapy Association (APTA) [50], and the Ottawa Panel [46]. Lastly, six guidelines targeting a mixed population of OA were developed by the Royal Dutch Society for Physical Therapy (KNGF) [48], the Osteoarthritis Research Society International (OARSI) [42], the American College of Rheumatology (ACR) [41], the European Alliance of Associations for Rheumatology (EULAR) [40], the Pan-American League of Associations for Rheumatology (PANLAR) [47], and the Royal Australian College of General Practitioners (RACGP) [49]. Table 2 presents the characteristics of the included guidelines.

### 3.3. Quality Assessment Module 

Table 3 and Table 4 present the quality assessment results of guidelines using the AGREE II tool for knee and hip OA, respectively. The overall quality scores ranged from 34% for the PANLAR guideline [47] to 76% for the KNGF guideline [48]. Three guidelines met our criteria for high-quality guidelines for both knee and hip OA: RACGP [49], AAOS [52], and KNGF [48].

To assess the content of these high-quality guidelines, we created matrices listing all clinical recommendations for knee and hip OA. For knee OA, a total of 100 recommendations were extracted and classified as non-pharmacological treatments mainly provided by HK physiotherapists (*n* = 39), non-pharmacological treatments partly provided by HK physiotherapists (*n* = 12), and other treatments that are not provided by HK physiotherapists or not within the scope of the practice (*n* = 49). For hip OA, a total of 79 recommendations were extracted and classified as non-pharmacological treatments mainly provided by HK physiotherapists (*n* = 27), non-pharmacological treatments partly provided by HK physiotherapists (*n* = 3), and other treatments that are not provided by HK physiotherapists or not within the scope of the practice (*n* = 49). Appendix A presents the results of the categorization for knee and hip OA, respectively. Appendix A presents the guidelines’ recommendations for knee and hip OA, respectively. 

Following the reviews and discussions of both the quality appraisal and the matrices, the adaptation panel voted (9 of 11 members, 82%) to include recommendations for knee OA treatments only from the RACGP [49] and AAOS [52] guidelines. In addition, the panel voted to modify the recommendations’ content of the exercises and modalities parameters (9/11, 82%). For hip OA, the panel voted (10/11, 91%) to include recommendations only from the RACGP [49] and AAOS [52] guidelines and excluded KNGF [48]. The panel voted to modify the recommendations’ content of the exercise and modalities parameters (10/11, 91%).

To modify the recommendations’ content, two reviewers (FAZ and CY) reviewed all the randomized controlled trials included in the RACGP [49] and AAOS [52] guidelines to extract their contents. The organizing committee decided to keep trials that were found to be clinically significant by these two guidelines’ developers. In addition, any recommendations that were based on a consensus among guideline panelists or used a non-randomized controlled trial design were discarded. Accordingly, the organizing committee rephrased the recommendations to reflect these criteria and add further details.

Appendix A presents summaries of trials that met these criteria for both knee and hip OA, respectively. In total, 28 and 20 recommendations for knee and hip OA were deemed suitable for adaptation, respectively. Table 5 and Table 6 present summaries of the key recommendations for the adapted CPG for the non-pharmacological treatments that can be provided by physiotherapists for knee and hip OA, respectively. 

Appendix A presents the differences between the original recommendations suggested by the RACGP [49] and AAOS [52] guidelines and our modified recommendations for knee and hip OA, respectively. 

### 3.4. Phase 3: Finalization

The final draft of the adapted guideline was sent to eight external review panels, representing the audience for this guideline. We also sent the traditional Chinese-translated recommendations to five patients with knee OA and one with hip OA. The comments and feedback arising from the panel and patients were reviewed and discussed by the panel members. Where relevant, clarifications were made to the recommendations and future implementation plans. 

The adapted guideline, all its pertained documents, a summary of recommendations, and a translation of the recommendations into the traditional Chinese language will be accessible on the official website of the HK Physiotherapy Association: https://www.hongkongpa.com.hk/, accessed on 29 October 2023.

### 3.5. Plan for Updating the Guideline

The adaptation panel plans to update the adapted guideline five years after publication. 

## 4. Discussion

This study adapted the published guidelines for the non-pharmacological treatments for knee and hip OA management that can be recommended to physiotherapists in HK. Our systematic search identified a total of 12 guidelines, of which three were of high quality [48,49,52]. With the availability of high-quality guidelines in the literature, the need to adapt them becomes more practical than developing new ones [15]. 

### 4.1. Similarities and Differences with Recommendations from the Original CPGs

For knee OA, the current guideline strongly recommends supervised and unsupervised land-based exercise (e.g., walking, muscle-strengthening exercise, and Tai Chi), aquatic exercise, patient education, and self-management. We moderately recommend neuromuscular training (i.e., balance, agility, coordination), weight management, canes, and braces. We conditionally recommend the following treatments: yoga, aquatic stationary cycling, massage therapy, manual therapy, transcutaneous electrical stimulation, wearable pulsed electromagnetic field devices, percutaneous electrical nerve stimulation, laser therapy, extracorporeal shockwave therapy, acupuncture, and heat. Generally, all recent guidelines share similar recommendations to ours, including patient education, weight management, supervised and unsupervised exercises, and reducing the loading on the knee using canes for patients with knee OA [54,55]. Our recommendations were generally identical to the ones recommended by either one of the AAOS [52] or RACGP [49]. If a recommendation was only synthesized by one of the two guidelines but not the other, then we relied on it to design our recommendation. An example of that is yoga, which was conditionally recommended by RACGP [49] but not discussed by AAOS [52]. If both guidelines synthesized a similar recommendation, our team would accept the recommendation from both guidelines. For instance, both of them highly recommend land-based exercises. If both guidelines synthesized contradicting recommendations, our team looked at the trials used by each guideline to synthesize relevant recommendations. Most of the time, we accepted the AAOS [52] recommendations as they included more recent and high-quality trials than RACGP [49]. For example, AAOS [52] highly recommended providing self-management education, while RACGP [49] was unable to provide recommendations/suggestions on it. We relied on the AAOS recommendation as it included recent moderate-to-high-quality trials [56,57,58], compared to the older and very low-quality trials [59] included by RACGP. 

For hip OA, the current guideline strongly recommends both supervised and unsupervised land-based exercise (e.g., walking, muscle-strengthening exercise, and Tai Chi) and aquatic exercise. We moderately recommend weight-loss management for overweight and obese patients. We conditionally recommend supervised aquatic strengthening exercises, manual therapy, cognitive-behavioral therapy, and assistive walking devices such as canes. 

Our recommendations were generally based on reviewing the ones recommended by the KNGF [48] and RACGP [49]. Although it was published in 2018 compared to KNGF (published in 2020), we relied more on recommendations from RACGP as they conducted systematic searches for each research question, while KNGF searched previously published systematic reviews first. In the event that these reviews were not available, they investigated trials or textbooks. A lack of extensive literature searches was the major limitation of the KNGF [48], as declared by the developers, which may have influenced the strength of the recommendations. 

On the other hand, our reliance on RACGP recommendations was not blind. As we set out our criteria, we mainly relied on evidence obtained from high-quality randomized controlled trials. Nonetheless, the content and strength of our recommendations were comparable to those of the RACGP, with the exception of weight management, transcutaneous electrical nerve stimulation, massage therapy, heat therapy, and transcutaneous electrical stimulation. Even though there are no published trials on weight management, our panel agreed that weight management is essential for overweight and obese individuals. The majority of osteoarthritis guidelines concur on the importance of weight management [60]. However, because of the lack of trials for hip OA, we lowered the strength of the recommendation to moderately recommended. 

As our panel members suggested modifying the recommendations by providing more details on the interventions’ parameters, we added remarks and statements to the original recommendations. We expect that this will boost the use of the recommendation among local physiotherapists, as the suggestion came from local expert clinicians who are familiar with the local needs and preferred language to design recommendations. This consideration of the trans-contextual issues of the local stakeholders is the core concept of adapting CPGs [16].

### 4.2. Stakeholder Considerations

As we recommend multimodal management for patients with knee and hip OA, it is important for individual physiotherapists to select the appropriate combination of therapies by considering their effectiveness (i.e., on pain levels and function), any potential risks to the patient (e.g., the suitability of the exercise to the age of the patient), and costs (e.g., the ability of the patient to bear the cost of yoga or Tai Chi classes). In addition to that, the physiotherapist shall consider the patient’s satisfaction with the provided treatment over time [61]. Current evidence provided by this guideline supports non-pharmacological interventions that can be provided by any physiotherapist in all HK health sectors: hospital authorities, non-governmental organizations, and private clinics. Yoga and Tai Chi may be the only two exceptions, as the government may not financially cover them. In addition, they may require a certified registered physiotherapist to provide them to patients. 

### 4.3. Cultural and Clinical Context

The cultural adaptation of clinical practice guidelines can significantly impact the recommendations for physiotherapists managing patients with knee or hip OA. By acknowledging and accounting for cultural and clinical contexts, physiotherapists can tailor their recommendations to align with the cultural values, preferences, and expectations of their patients in HK, resulting in more effective and patient-centered care.

In HK’s cultural context, there are several cultural factors that can impact the acceptance and adherence to recommended treatments [62]. These include traditional health beliefs, family involvement, language and communication, and socioeconomic and environmental factors. Physiotherapists need to take cultural nuances into account and address them in order to ensure that the recommendations are meaningful and relevant to the local population. By implementing these practices, it fosters a strong sense of trust between physiotherapists and their patients. This, in turn, promotes active participation from patients, leading to better treatment results [14,15].

Regarding traditional health beliefs, individuals in HK prefer a comprehensive approach that integrates complementary and alternative therapies with conventional physiotherapy interventions [63]. Physiotherapists have the option to incorporate traditional Chinese medicine techniques, such as acupuncture, into the treatment plan to accommodate patients’ cultural preferences. Due to its perceived safety compared to Western medicine, traditional Chinese medicine enjoys high regard among the general public in HK [64], with a large majority of patients with low back pain opting for complementary and alternative therapies such as traditional Chinese medicine and acupuncture [65]. While most physiotherapists in HK have received comprehensive training in complementary and alternative therapies, such as acupuncture, as part of their undergraduate education, those who obtained their degrees from universities overseas may require further training to effectively address patients’ needs in the local community.

Additionally, acknowledging the significance of family involvement in decision-making and considering the support systems within the cultural context can help optimize treatment outcomes. The interdependence and connectedness of family members are deeply rooted in Chinese Confucian beliefs [66]. Language and communication play a vital role in cultural contexts. Translating and adapting the recommendations into the local language, along with using culturally appropriate examples and analogies during patient education, can enhance comprehension and foster patient cooperation [67].

Furthermore, socioeconomic and environmental factors influenced by culture can affect the feasibility and accessibility of treatments [68,69]. Physiotherapists should consider these when tailoring recommendations for patients. For example, suggesting exercises that can be performed in small living spaces, considering financial constraints when recommending assistive devices, or highlighting community resources for ongoing support and follow-up care can all enhance the cultural appropriateness of the recommendations.

In HK’s clinical context, the cultural adaptation of CPGs should consider the applicability of recommendations at the policy level [14,15]. While patients in Western countries have direct access to physiotherapy services, the traditional hierarchical structure in HK requires patients with knee and hip OA to obtain a medical referral [70]. Soon, however, patients will be able to access physiotherapy services directly (i.e., without a medical referral) [71]. While this is a significant professional development for physiotherapists, it will increase their pressure to deliver evidence-based practice. The objective of the present study was to assist healthcare professionals in effectively managing patients diagnosed with knee and hip OA. At the request of our expert clinicians, we have revised the recommendations to incorporate parameters for non-pharmacological modalities. This will help ensure that the recommendations are easily understood and can be implemented effectively.

Another concern is the shortage of physiotherapists in HK, which is impacting the availability of services for older individuals. Specifically, 34% of the 174 elderly services units are unable to fill their open vacancies [72]. There is an increasing demand for physiotherapists to see a higher number of patients per day in public settings, which could potentially impact the quality of healthcare services provided [73]. Furthermore, due to the shortage of physiotherapists, patients with knee OA who consult a general practitioner and receive a referral to see a physiotherapist may experience significant delays in accessing their services, with wait times of up to 6 months in public settings. Hence, physiotherapists may need to explore the implementation of group-based supervised interventions as a means to effectively manage a larger number of patients while maintaining close monitoring of their progress.

### 4.4. Dissemination and Implementation Plan

A 2019 systematic review revealed that many interventions provided by physiotherapists to manage musculoskeletal conditions, including knee OA, were not recommended [74]. This trend has not changed over the past three decades, with almost 40% of the provided physiotherapy treatments not recommended [75]. There are several individual (lack of knowledge, lack of English proficiency, lack of skills about research and statistics, lack of interest) and organizational (lack of time, lack of access, lack of resources, and generalizability of the recommendations to other contexts) factors that impede the use of recommended care in clinical practice among physiotherapists [76]. Knowledge translation is a process used to tackle such factors and bridge the gap between recommended care and clinical practice [77]. The closure of this gap is possible by developing and implementing knowledge translation strategies targeting physiotherapists, patients with knee and hip OA, and healthcare organizations [78]. Thus, to facilitate the implementation of our guideline, we considered the Guideline Implementation Planning Checklist [78]. To increase the local physiotherapists’ awareness, the HK Physiotherapy Association will endorse this guideline and help disseminate its content and resources via its website (https://www.hongkongpa.com.hk/, accessed on 29 October 2023). The implementation tools include handouts for patients with knee and hip OA and physiotherapists (Table 5 and Table 6), algorithms (Figure 2 and Figure 3), regional and local conferences [79], and seminars.

The ADAPTE framework and other frameworks are limited in providing a dissemination and implementation methodology to allow for the uptake of the adapted recommendations into the local context. Therefore, we recommend developing post-hoc, clear, and detailed methods to implement the adapted guidelines. In addition, although the ADAPTE framework provides a sample survey to seek the external review committee’s feedback on the adapted guideline (Tool 17), we have modified this tool to meet our needs (see Appendix A). However, there is no tool to seek the patients’ feedback on the recommendations. To overcome this issue, we have developed our own tool to achieve this mission (see Appendix A). 

### 4.5. Future Research 

There are several frameworks that may be used to adapt CPGs to the local context [80]. A recent systematic review identified eight frameworks [80], of which half—RAPADAPTE, Adapted ADAPTE, MAGIC, and CAN-IMPLEMENT—were derived from the ADAPTE framework. These adaptation frameworks consider the trans-contextual issues of the local settings for the adapting countries [16]. This is of paramount importance as it is considered the cornerstone to facilitate the guideline implementation [81].

The ADAPTE framework has been utilized across different medical conditions [80] in both developed [17,18,19,20,21,22,23,24,25,26] and developing [27,28,29,30,31,32] countries. Importantly, the ADAPTE framework considers the local needs, medical priorities, different policies, and availability of resources for the targeted country/setting/organization [80]. Furthermore, it comes with a detailed resource toolkit and a user manual [33].

In our opinion, the ADAPTE framework has a rigorously well-developed, structured, and resilient approach to handling the differences between developing and adapting contexts. Nevertheless, we found that it needs to be modified in some of its steps or tools. For instance, in the setup phase, the order of the steps was to check the feasibility of the adaptation, establish a committee, and then select a topic. However, the logical way should be to select the topic first and then check its adaptation feasibility. Furthermore, some of the tools need amendments or further expansion. For example, although Tool 2 provides a list of guideline clearinghouses and resources to find CPGs, most of these clearinghouses are either inaccessible or have not been updated for a long time due to insufficient funding. In addition, these resources mainly focus on the medical literature rather than other healthcare disciplines, such as physiotherapy. These issues have also been raised by other adaptation studies using the ADAPTE framework [17,27,28].

Despite these limitations, the ADAPTE framework developers allowed their users to either use, ignore, or modify the original steps or tools. Further, most of the prior adaptation studies that used the ADPATE framework benefited from these flexibility options, mostly by modifying the tools and/or steps [27,28,29,30] and being less likely to use them as is [26,32]. Therefore, we have proposed some major and minor modifications to the ADAPTE framework to facilitate and shorten the adaptation process (see Appendix A). This practicability and friendliness of the ADPATE framework may explain its popularity over other frameworks [80].

Our adaptation panel members were clinically skilled and methodologically experts in the field of knee and hip OA. This enriched the discussion and feedback from the panel members, which allowed the modification of the content and changes to the recommendations to suit local needs. This is another feature suggested by the ADAPTE framework: using a multidisciplinary team. However, the framework does not provide any definition of the meaning of ‘experts,’ their level of expertise, or how to choose them among other peers. This concern has also been reported previously by other colleagues [17], recommending further refinements on future updates of the ADPTE framework. We agree with this recommendation and suggest further distinguishing between decision-makers and only skilled ‘experts’.

Research that uncovers the practice patterns of local physiotherapists to explore how they manage patients with knee and hip OA is needed. This will help explore the gaps in practice compared to recommended care. This, in turn, will allow for the design of a knowledge translation strategy that can be tailored to the local needs of physiotherapists [78].

### 4.6. Strengths and Weaknesses

There are several strengths in our work. First, use a comprehensive search covering the main electronic databases and clearinghouse websites to identify potential guidelines. Second, using the ADAPTE framework [15] over the other frameworks provided a clear path with useful resources to adapt guidelines on knee and hip OA. Third, modify the recommendations to provide more clinically useful recommendations by only including interventions with parameters proven to be effective from clinical trials. As suggested by local senior clinicians, this is important to allow the recommendations to be implementable and more trustworthy, as it shows the reliance on a high level of evidence. Fourth, although this is our first experience with adaptation studies, the adaptation process (see Appendix A) took around 25 months, which is considered to be comparable with the development of a de novo guideline (~2–3 years) [82]. Other studies that used the ADAPTE framework reported different durations to accomplish their projects, ranging from 6 months [83] to 36 months [28].

Nevertheless, our study has some limitations. First, our search was limited to guidelines published in English or Chinese. We identified several guidelines published in other languages, which might have limited the comprehensiveness of our recommendations. Second, the AGREE II tool has some limitations related to its assessment of clinical credibility. The AGREE II developers have also developed AGREE-REX: Recommendation Excellence [84], which was designed to evaluate the clinical credibility and implementability of the guidelines’ recommendations. We have not used AGREE-REX, as it is not mentioned in the ADAPTE framework. Instead, the ADAPTE framework relies on the discussion within the adaptation panel. We plan to include AGREE-REX in our future adaptation studies to evaluate its usefulness. Third, we found that the ADAPTE framework was lengthy and required thorough training among its users. This observation has been described by almost all prior users of the ADAPTE framework [17,18,27,29,30,32,83].

## 5. Conclusions

This is the first study to provide a detailed description of the use of the ADAPTE framework in physiotherapy research. We found the ADAPTE framework to be efficient, well-structured, and rigorously guided in the cross-cultural adaptation of guidelines to the local context. Furthermore, this framework allows modifications to be applied to the adaptation process according to contextual needs [15]. This flexibility allowed us to add further steps to modify the recommendations based on our panel’s preferences and needs. 

Current evidence on the effectiveness of conservative management of knee and hip OA suggests the use of all forms of supervised or unsupervised exercises. Considering patient preferences and the availability of resources, physiotherapists can offer a multimodal intervention to patients with knee or hip OA. A continual monitoring of the outcomes of the patients with knee and hip OA (pain intensity and functional disability) should regularly guide the physiotherapist about the progress of the provided interventions. 

## Figures and Tables

**Figure 1 healthcare-11-02964-f001:**
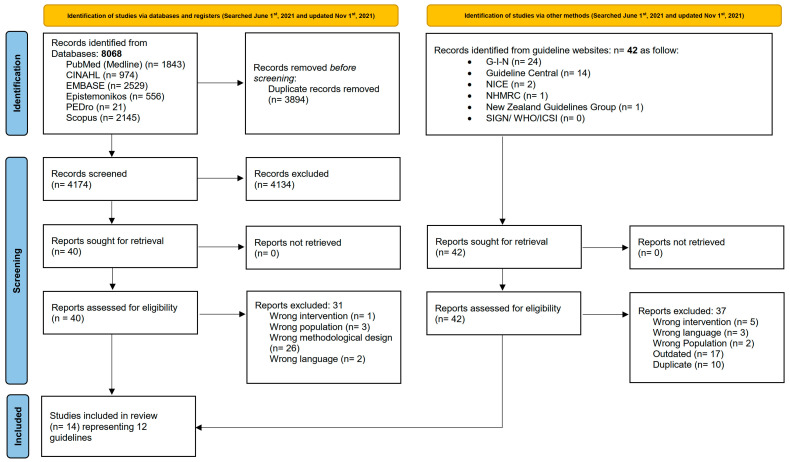
PRISMA 2020 flow diagram for new systematic reviews, which included searches of databases, registers, and other sources [53].

**Figure 2 healthcare-11-02964-f002:**
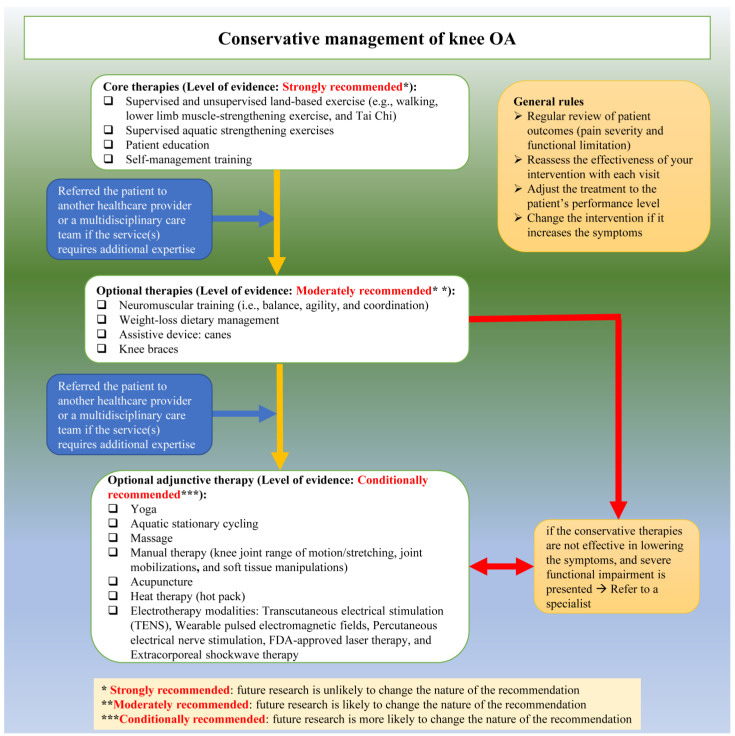
Conservative management of knee OA algorithms.

**Figure 3 healthcare-11-02964-f003:**
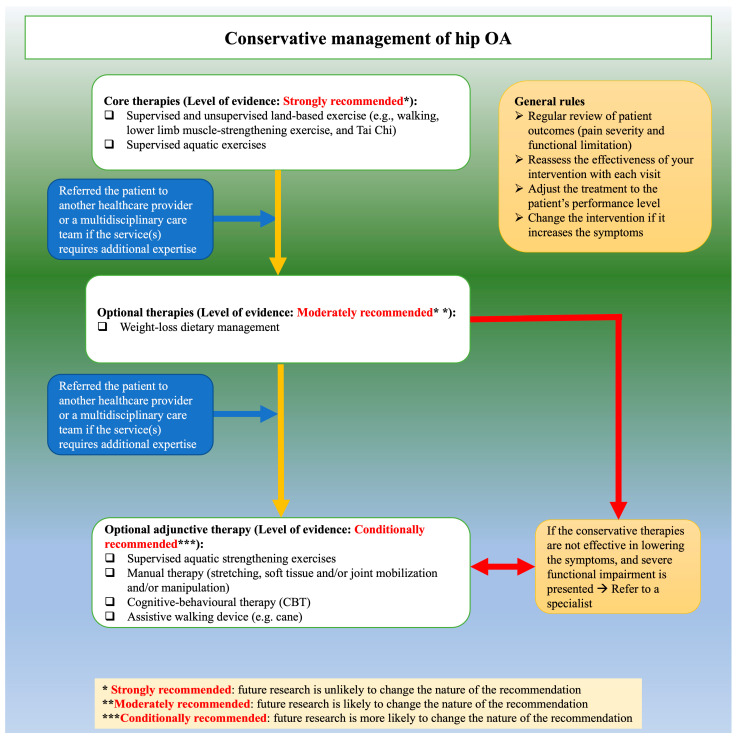
Conservative management of hip OA algorithms.

**Table 1 healthcare-11-02964-t001:** A list of the medical databases and guideline clearinghouses that have been searched.

#	Name of the Database	Country	Website
Medical databases			
i	MEDLINE (PubMed)	USA	https://www.nlm.nih.gov/medline/medline_overview.html, accessed on 23 March 2021.
ii	CINAHL Complete (EbscoHost)	USA	https://www.ebsco.com/products/research-databases/cinahl-database, accessed on 23 March 2021
iii	EMBASE	The Netherlands	https://www.embase.com/, accessed on 23 March 2021
iv	Epistemonikos	Chile	https://www.epistemonikos.org/, accessed on 23 March 2021
v	Scopus	International	https://www.scopus.com/home.uri, accessed on 23 March 2021
vi	PEDro	Australia	https://pedro.org.au/, accessed on 23 March 2021
Guideline Clearinghouse			
i	Guidelines International Network (G-I-N)	International	http://www.g-i-n.net, accessed on 23 March 2021
ii	Guideline Central	International	https://www.guidelinecentral.com/, accessed on 23 March 2021
iii	National Institute for Health and Care Excellence (NICE)	UK	http://www.nice.org.uk/guidance/, accessed on 23 March 2021
iv	National Health and Medical Research Council (NHMRC)	Australia	http://www.nhmrc.gov.au/guidelines-publications, accessed on 23 March 2021
v	New Zealand Guidelines Group	New Zealand	https://www.health.govt.nz/, accessed on 23 March 2021
vi	World Health Organization (WHO)	International	https://www.who.int/publications/who-guidelines, accessed on 23 March 2021
vii	Scottish Intercollegiate Guidelines Network (SIGN)	UK	http://www.sign.ac.uk/guidelines/index.html, accessed on 23 March 2021
viii	Institute for Clinical Systems Improvement (ICSI)	USA	https://www.icsi.org/guidelines/, accessed on 23 March 2021

**Table 2 healthcare-11-02964-t002:** Characteristic of the included guidelines.

Developer	Location	Publication Date	End of Search Date	Type/Location of Arthritis
AAOS-Knee	USA	2021	28 April 2020	KOA
Dutch-KNGF	Netherlands	2020	December 2016–August 2017	KOA + HOA
OARSI	International	2019	July 2018	KOA + HOA + hand OA
ACR	USA	2019	August 2018	KOA + HOA + hand OA
Australian-RACGP	Australia	2018	December 2016	KOA + HOA
EULAR	Europe	2018	April 2017	KOA + HOA, RA, spondylarthritis
Turkish	Turkey	2018	Jane 2015	KOA
APTA	USA	2017	2016	HOA
OTTAWA Knee	Canada	2017	May 2016	KOA
AAOS-Hip	USA	2017	March–April 2016	HOA
PANLAR	South America	2016	2014	KOA + HOA + hand OA
OTTAWA Hip	Canada	2016	May 2015	HOA

OA: osteoarthritis; KOA: knee osteoarthritis; HOA: hip osteoarthritis; RA: rheumatoid arthritis.

**Table 3 healthcare-11-02964-t003:** Domain and total scores of the knee osteoarthritis guidelines using the AGREE II tool.

Guideline	Domain 1. Scope and Purpose	Domain 2. Stakeholder Involvement	Domain 3. Rigor of Devlopment	Domain 4. Clarity of Presentation	Domain 5. Applicability	Domain 6. Editorial Independence	Overall Quality Score
KNGF	97%	94%	76%	97%	44%	50%	76%
RACGP	100%	61%	86%	89%	23%	42%	67%
AAOS-Knee	94%	64%	90%	92%	17%	46%	67%
OARSI	81%	78%	60%	81%	4%	92%	66%
APTA	69%	67%	65%	100%	27%	58%	64%
ACR	89%	86%	68%	83%	4%	25%	59%
OTTAWA-Knee	75%	67%	49%	83%	4%	33%	52%
EULAR	86%	83%	50%	81%	25%	25%	58%
TLAR	56%	33%	46%	78%	0%	17%	38%
PANLAR	39%	47%	21%	78%	2%	17%	34%

Guidelines were included if four out of six domains had scores > 60% and domain three had scores > 75%. KNGF: Royal Dutch Society for Physical Therapy; RACGP: Royal Australian College of General Practitioners; AAOS-Knee: American Academy of Orthopaedic Surgeons guideline for knee OA; OARSI: Osteoarthritis Research Society International; APTA: American Physical Therapy Association; ACR: American College of Rheumatology; OTTAWA-Knee: Ottawa guideline for knee OA; EULAR: European Alliance of Associations for Rheumatology; TLAR: Turkish League Against Rheumatism; PANLAR: Pan-American League of Associations for Rheumatology.

**Table 4 healthcare-11-02964-t004:** Domain and total scores of the hip osteoarthritis guidelines using the AGREE II tool.

Guideline	Domain 1. Scope and Purpose	Domain 2. Stakeholder Involvement	Domain 3. Rigor of Devlopment	Domain 4. Clarity of Presentation	Domain 5. Applicability	Domain 6. Editorial Independence	Overall Quality Score
KNGF	97%	94%	76%	97%	44%	50%	76%
RACGP	100%	61%	86%	89%	23%	42%	67%
AAOS-Hip	83%	75%	92%	72%	50%	75%	75%
OARSI	81%	78%	60%	81%	4%	92%	66%
APTA	69%	67%	65%	100%	27%	58%	64%
ACR	89%	86%	68%	83%	4%	25%	59%
OTTAWA-Hip	81%	47%	55%	78%	4%	29%	49%
EULAR	86%	83%	50%	81%	25%	25%	58%
PANLAR	39%	47%	21%	78%	2%	17%	34%

Guidelines were included if four out of six domains had scores > 60% and domain 3 had scores > 75%. KNGF: Royal Dutch Society for Physical Therapy; RACGP: Royal Australian College of General Practitioners; AAOS-Hip: American Academy of Orthopaedic Surgeons guideline for hip OA; OARSI: Osteoarthritis Research Society International; APTA: American Physical Therapy Association; ACR: American College of Rheumatology; OTTAWA-Hip: Ottawa guideline for hip OA; EULAR: European Alliance of Associations for Rheumatology; PANLAR: Pan-American League of Associations for Rheumatology.

**Table 5 healthcare-11-02964-t005:** Summary of the key recommendations for the adapted clinical practice guideline for the non-pharmacological treatments that can be provided by physiotherapists for knee OA.

*******	We strongly recommend supervised and unsupervised land-based exercise (e.g., walking, muscle-strengthening exercise, and Tai Chi) and/or aquatic exercises to improve pain and function among patients with knee OA. Remarks: All types of exercise were found to be significantly better than no exercise. However, the results were too mixed to determine which exercise program was superior. The exercise program should last for at least 6 weeks, and physiotherapists can use the frequency, intensity, time, and type (FITT) principle to prescribe exercises for individual patients.
*******	We strongly recommend supervised aquatic strengthening exercises to improve pain and function for patients with knee OA. Remarks: The recommended program consists of 30 min of supervised aquatic strengthening exercises, preceded by a 5-min warm-up and followed by a 5-min cool-down, twice a week for 6 weeks.
*******	We strongly recommend providing patient education to patients with knee OA as a means to reduce pain and improve function. Remarks: Patient education can be delivered through various modes, such as an educational pamphlet, a video, and one to several days of education per month. The content of the education could involve various forms of exercise, proven effective interventions, and self-management techniques for knee OA, including pain management, medication compliance, and stress management.
*******	We strongly recommend self-management training to improve pain and function for patients with knee OA in both the short and long term. Remarks: Self-management training should cover pain coping skills training, exercises, and behavioral weight management and should be provided to patients once a week for at least 6 weeks, with each session lasting at least 60 min.
******	We moderately recommend providing neuromuscular training programs that include balance, agility, and coordination exercises, in addition to traditional exercises, to improve functions such as walking speed and balance for patients with knee OA. Remarks: Kinaesthesia and balance exercises (e.g., retro-walking, walking on toes, leaning to the sides, balance-board exercises, mini-trampoline exercises, plyometric exercises, etc.) combined with traditional strengthening exercises should be conducted three times a week for 8 weeks.
******	We moderately recommend weight-loss dietary management combined with exercises to reduce pain and improve function for overweight and obese patients with knee OA. Remarks: Physiotherapists should encourage overweight (BMI ≥ 25 kg/m^2^) or obese (BMI ≥ 30 kg/m^2^) patients with knee OA to follow a weight-loss program to lose at least 5% of their body weight. The dietary program should be combined with exercise.
******	We moderately recommend using canes to reduce pain and improve function for patients with knee OA, if indicated. Remarks: Wooden canes with a T-shaped handle can be used for patients with knee OA.
******	We moderately recommend Knee braces can be used to reduce pain, improve function, and enhance the quality of life for patients with knee OA. Remarks: The Bioskin Patellar Tracking Q Brace (worn for as long as tolerated per day for 6 weeks) or the REBEL RELIEVER unloading knee brace (worn for at least 6 h/day for 6 weeks) can be used for patients with knee OA.
*****	We conditionally recommend yoga to reduce pain and improve mobility in patients with knee OA. Remarks: Supervised yoga can be prescribed for 40 min per day over a period of 2 weeks. After the supervised sessions, patients should be advised to continue with 40-min yoga sessions at home for the next 10 weeks. The yoga program could include shithilikarana vyayamas or sakti vikasaka, followed by yoga asanas and relaxation techniques.
*****	We conditionally recommend aquatic stationary cycling to improve function for some patients with knee OA. Remarks: Supervised (for a maximum of 4 patients), aquatic cycling should last for 45 min twice a week for 12 weeks.
*****	We conditionally recommend massage therapy combined with usual care to reduce pain and improve function for patients with knee OA. Remarks: A 60-min total body massage could be offered once a week for 8 weeks, or effleurage and petrissage techniques could be applied to the knee joint in the direction of lymph drainage for 15–20 min, twice a week for 3 weeks.
*****	We conditionally recommend manual therapy in combination with a standardized knee exercise program to reduce pain and improve function for patients with knee OA. This should be considered only as an adjunctive treatment to enable engagement with active management. Remarks: Manual therapy may include knee accessory joint mobilizations, knee joint range of motion/stretching, and soft tissue manipulations of the quadriceps, rectus femoris, hamstring, and gastrocnemius muscles twice a week for a period of 4 weeks as an adjunctive treatment.
*****	Transcutaneous electrical stimulation might be used as an adjunctive treatment to reduce pain and improve function in patients with knee OA. Remarks: Patients can use the device as much as needed using four electrodes around the knee joint line (two medially and two laterally) in continuous mode (program A: 110 Hz, 50 μs). All electrical pulses should be asymmetric and biphasic for 30 min, up to 6 weeks.
*****	We conditionally recommend using a wearable pulsed electromagnetic field device to reduce pain and improve function for patients with knee OA. Remarks: A wearable pulsed radiofrequency energy device (ActiPatch) can be used as adjunctive therapy. We suggest the following parameters for 12 h/day for 4 weeks: carrier frequency at 27.12 MHz; 1000 Hz pulse rate; 100 μs burst width; and peak burst output power ∼0.0098 W/surface area of ∼103 cm^2^.
*****	We conditionally recommend percutaneous electrical nerve stimulation to reduce pain and improve function for patients with chronic knee OA. Remarks: Percutaneous electrical nerve stimulation could be used as an adjunctive therapy. We suggest using the following parameters for 20 min/day, three times/day for 8 weeks: 2–6 Hz for frequency and 150 ms for pulses.
*****	We conditionally recommend FDA-approved laser therapy to reduce pain and improve function for patients with knee OA. Remarks: Laser therapy can be used as an adjunctive therapy. We suggest either using (a) a 5-min stimulation time, 200-nanosecond maximum pulse duration, 2.5 kHz pulse frequency, 20 W maximum output/pulse, 10 mW average power, 1 cm^2^ surface, 3 J total energy, and 30 J accumulated dose, five times a week for 2 weeks; or (b) a Neodymium:Yttrium–Aluminum–Garnet (Nd:YAG) high-intensity laser therapy with 1064 nm wavelength on the medial and lateral sides of the knee joint line for 8 min, at a frequency of 30 Hz with a peak power of 5 W, a duty cycle of 70%, energy density of 60 J/cm^2^, and total energy of 2400 J/session, three times a week for 4 weeks.
*****	We conditionally recommend extracorporeal shockwave therapy to reduce pain and improve function in patients with knee OA. Remarks: Extracorporeal shockwave therapy could be used as an adjunctive therapy. The parameters of therapy may include (a) 2000 pulses of 8-Hz frequency at 2.5 bars of pneumatic pressure, once a week for 4 weeks; (b) 4000 pulses at 0.25 mJ/mm^2^ and a frequency of 6 Hz/s, once a week for 12 weeks; or (c) 2500 pulses at a pressure of 3 bars and a frequency of 12 Hz, twice a week for 5 weeks.
*****	We conditionally recommend acupuncture to improve pain and function. Remarks: Acupuncture can be accompanied by an electro-stimulator for an average of 8 weeks, twice a week for 20–30 min, using different acupuncture points.
*****	We conditionally recommend heat therapy, such as using a hot pack as an adjunctive therapy or as part of the self-management home program, to reduce pain for patients with knee OA.
**?**	Due to a lack of evidence, the committee decided not to make any recommendation/suggestion regarding the use of trigger point dry needling.
**?**	Due to a lack of evidence, the committee decided not to make any recommendation/suggestion regarding the use of patellar taping.
**?**	Due to a lack of evidence, the committee decided not to make any recommendation/suggestion regarding the use of shoe orthotics (medial wedge insoles, shock-absorbing insoles, and arch supports).
**?**	Due to a lack of evidence, the committee decided not to make any recommendation/suggestion regarding the use of shortwave therapy.
*****	We conditionally recommend against the provision of unloading shoes, minimalist footwear, or rocker-sole shoes for patients with knee OA. Instead, physiotherapists may advise patients with knee OA to use shock-absorbing footwear.
*****	We conditionally recommend against the provision of kinesiotaping for patients with knee OA.
*****	We conditionally recommend against the provision of cold therapy, such as using an ice pack, for patients with knee OA.
*****	We conditionally recommend against the provision of interferential therapy for patients with knee OA.
*******	We strongly recommend against the provision of shoe orthotics (strapped or lateral wedged insoles) for patients with knee OA.
*******	Strongly recommended: future research is unlikely to change the nature of the recommendation.
******	Moderately recommended: future research is likely to change the nature of the recommendation.
*****	Conditionally recommended: future research is more likely to change the nature of the recommendation.
**?**	Neutral: unable to recommend.
*****	Conditionally recommend against: future research is more likely to change the “against” nature of the recommendation.
******	Moderately recommend against: future research is likely to change the “against” nature of the recommendation.
*******	Strongly recommend against: future research is unlikely to change the “against” nature of the recommendation.

**Table 6 healthcare-11-02964-t006:** Summary of the key recommendations for the adapted clinical practice guideline for the non-pharmacological treatments that can be provided by physiotherapists for hip OA.

*******	We strongly recommend supervised and unsupervised land-based exercise (e.g., walking, muscle-strengthening exercise, and Tai Chi) and/or aquatic exercise to improve pain, function, and quality of life for patients with hip OA. Remarks: All types of exercises were found to be significantly better than no exercise. However, the results were too mixed to determine which exercise program was better than others. The exercise program should last at least 6 weeks. Physiotherapists can prescribe the exercises using the frequency, intensity, time, and type (FITT) principle.
******	We moderately recommend weight-loss management to reduce pain and improve function in patients with hip OA who are overweight or obese. Remarks: Physiotherapists should encourage overweight (BMI ≥ 25 kg/m^2^) or obese (BMI ≥ 30 kg/m^2^) patients with hip OA to follow a weight-loss program to lose at least 5% of their body weight. The dietary program should be combined with exercise.
*****	We conditionally recommend supervised aquatic strengthening exercises to improve pain, function, and quality of life for patients with hip OA. This will depend on individual preferences and the availability of pools in clinical settings. Remarks: The supervised aquatic strengthening exercises should last for 30–60 min, preceded by a 5-min warm-up and followed by a 5-min cool-down, 2–3 times a week for 6–12 weeks.
*****	We conditionally recommend manual therapy (stretching, soft tissue, and/or joint mobilization and/or manipulation) to improve pain, function, and quality of life for patients with hip OA. This should be considered only as an adjunctive treatment to enable engagement with active management. Remarks: Manual therapy may include trigger point release therapy, muscular and fascial stretching, and joint manipulations (thrust, non-thrust, distraction, anterior-posterior glide, or posterior-anterior glide), performed 1–2 times per week for 6 weeks. This should only be considered an adjunctive treatment.
*****	We conditionally recommend cognitive-behavioral therapy (CBT) combined with exercises to improve pain and function among patients with hip OA. Remarks: CBT may include relaxation techniques, pleasant imagery, pain coping skills training, and problem-solving techniques, with sessions lasting 35–45 min per week for 8 weeks. CBT may be provided in person or via online programs.
*****	We conditionally recommend assistive walking devices such as canes be used for patients with hip OA, depending on their individual preferences and capabilities.
**?**	Due to a lack of evidence, the committee decided not to recommend/suggest self-management. However, physiotherapists should educate patients about the condition they manage, including its optimal care and prognosis.
**?**	Due to a lack of evidence, the committee decided not to recommend/suggest the use of transcutaneous electrical stimulation (TENS).
**?**	Due to a lack of evidence, the committee decided not to recommend/suggest the use of shoe orthotics.
**?**	Due to a lack of evidence, the committee decided not to recommend/suggest the use of massage therapy for patients with hip OA.
**?**	Due to a lack of evidence, the committee decided not to recommend/suggest the use of pulsed electromagnetic therapy for patients with hip OA.
**?**	Due to a lack of evidence, the committee decided not to recommend/suggest the use of shortwave therapy for patients with hip OA.
**?**	Due to a lack of evidence, the committee decided not to recommend/suggest the use of therapeutic heat therapy (e.g., hot packs) for patients with hip OA.
*****	We conditionally recommend against the use of laser therapy for patients with hip OA.
*****	We conditionally recommend against the use of extracorporeal shockwave therapy for patients with hip OA.
*****	We conditionally recommend against the use of interferential therapy for patients with hip OA.
*****	We conditionally recommend against the use of therapeutic ultrasound for patients with hip OA.
*****	We conditionally recommend against the use of local cold applications (e.g., ice packs) for patients with hip OA.
*****	We conditionally recommend against the use of kinesiotaping for patients with hip OA.
*****	We conditionally recommend against the use of acupuncture for patients with hip OA.
*******	Strongly recommended: future research is unlikely to change the nature of the recommendation.
******	Moderately recommended: future research is likely to change the nature of the recommendation.
*****	Conditionally recommended: future research is more likely to change the nature of the recommendation.
**?**	Neutral: unable to recommend.
*****	Conditionally recommend against: future research is more likely to change the “against” nature of the recommendation.
******	Moderately recommend against: future research is likely to change the “against” nature of the recommendation.
*******	Strongly recommend against: future research is unlikely to change the “against” nature of the recommendation.

## Data Availability

The data used and analyzed during the current study are available from the corresponding author upon reasonable request.

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
