# Peer review of "Adapting a Clinical Practice Guideline for Management of Patients with Knee and Hip Osteoarthritis by Hong Kong Physiotherapists"

_healthcare, 2023, doi:10.3390/healthcare11222964_

Round 1

Reviewer 1 Report (Previous Reviewer 1)

Comments and Suggestions for Authors

Thank you for improving the manuscript. Taking into account the cultural context is important here.

Author Response

Dear reviewer,

We sincerely thank you for the insightful comments on our submitted manuscript.

All the best. 

Reviewer 2 Report (Previous Reviewer 2)

Comments and Suggestions for Authors

The authors have completed a detailed revision of their submission based upon feedback provided during the review process. The addition/expansion of section 4.3 in the discussion is a beneficial element to the paper which speaks top the central them of the paper in adapting guidelines to the local/cultural population of interest. The authors are also actively utilizing the process to provide recommendations and help refine and develop the ADAPTE process. Each of the 44 questions/recommendations provided were addressed and either adopted, discussed or deferred based upon further discussion showing the active development, autonomy and also considered review by the authors. At this point I am not seeing any need for further revision though defining manual therapy the same way in both charts would be a nice final revision.

Author Response

Dear reviewer,

We sincerely thank you for the insightful comments on our submitted manuscript. 

Thanks for suggesting improving the presentation of Fig 2 by adding a definition for manual therapy. We have added that (see page 22- line 476).

All the best. 

Reviewer 3 Report (Previous Reviewer 4)

Comments and Suggestions for Authors

All points of the paper requested from the authors have been resolved

Author Response

Dear reviewer,

We sincerely thank you for the insightful comments on our submitted manuscript.

All the best. 

This manuscript is a resubmission of an earlier submission. The following is a list of the peer review reports and author responses from that submission.

Round 1

Reviewer 1 Report

Comments and Suggestions for Authors

Thank you very much for the opportunity to review an interesting article. However, I have a few comments:

1. Please provide key words according to MeSH

2. What are the chances of introducing these guidelines into clinical practice?

Author Response

Dear reviewer,

We sincerely appreciate your insightful comments on this manuscript. Please see the attached file for our responses to your comments and/or revisions.

Sincere regards,

Research team

Reviewer 2 Report

Comments and Suggestions for Authors

I have provided 41 comments to address in the attached PDF. These ideally are not difficult to address, and will assist in improving the clarity of the paper. A key issue is a consistent use of the title physiotherapist and the term physiotherapy. This is described in the comments. A few points of clarification are suggested.

Comments on the Quality of English Language

These are address with a few word edits in the PDF document and comments.

Author Response

(The authors gave the same response as above.)

Reviewer 3 Report

Comments and Suggestions for Authors

Dear Authors, 

thank you for providing the article on "Adapting clinical practice guideline for patients with knee and hip osteoarthritis to Hong Kong physiotherapists". 

Guidline Adaptation is a powerful tool for getting research into practice. 

Still, even if there is a need to justify the usage of ADAPTE framework, there is no need for citation 17-32 for justification this framework is sometimes used. Please reduce to 2-3 citations.

Unfortunately this figures (like figure 2 or 3) are of such poor quality that I can hardly read it. Please revise.

Good luck and well done!

Author Response

(The authors gave the same response as above.)

Reviewer 4 Report

Comments and Suggestions for Authors

This review, which adapts the clinical practice guideline for patients with knee and hip osteoarthritis to Hong Kong physiotherapists, is very interesting. However, I would like to suggest some questions:

Have the factors involved in pain management in relation to gender been taken into account?

The most important sociocultural values in China are guanxi, mianzi, personal trust (xinren), renqing, harmony, hierarchy, and collectivism. Have they been taken into account for pain assessment?

The acronym for Osteoarthritis Research Society International is OARSI, not ORASI (misspelled in tables 2, 3 and 4).

Some bibliographical references are not well written, such as 19, in which two surnames appear for the authors.

Author Response

(The authors gave the same response as above.)

Round 2

Reviewer 1 Report

Comments and Suggestions for Authors

The authors took into account all the reviewers' comments, but I still believe that the article has little scientific significance because it only concerns guidelines for the Hong Kong population

Author Response

Dear Reviewer,

We appreciate your assessment of our work and the concern you raised regarding the article's relevance only to the Hong Kong population. We contend, however, that this particular regional focus does not detract from the work's broader scientific impact or its potential to inform practice in other places, which we elaborate on below.

Firstly, the unique focus of our study on Hong Kong is not a limitation but an essential element of effective implementation science. By tailoring our strategy to this distinct socio-cultural and regulatory environment, we provide insights suitable for and immediately applicable to this population. While we acknowledge that our study’s practical implications might be directly relevant to Hong Kong, the methodological applications aimed at refining adaptation processes can benefit various regions. It underscores the broader theme that tailoring strategies to local contexts is crucial for the wider applicability and overall effectiveness of public health initiatives.

Secondly, our work contributes to "methodological research" by highlighting essential cultural modifications to make the ADAPTE framework suitable for East Asian settings. It’s noteworthy that several methodological papers have been produced in various regions, all reinforcing the importance of regional customization of global health guidelines. Thus, while our study may appear only regionally pertinent, it feeds into a larger, globally relevant discourse, highlighting the universal need to customize health care practices.

Lastly, our study's methodology and findings will inform future research aimed at adapting clinical practice guidelines not only in Hong Kong but also in East Asia and beyond, leveraging cultural similarities and health policy commonalities.

While we appreciate your comment about the regional focus of our paper, we would like to assure you that our research findings could stimulate improvements in practice and discussion in the broader scientific community about the vital need for regional customization when implementing broad guidelines.

Best Regards,